# Vancomycin-Induced Organizing Pneumonia: A Case Report and Literature Review

**DOI:** 10.3390/medicina57060610

**Published:** 2021-06-11

**Authors:** Young-Shin Lee, Yu-Mi Lee

**Affiliations:** Division of Infectious Diseases, Department of Internal Medicine, Kyung Hee University Hospital, Kyung Hee University School of Medicine, 23, Kyungheedae-ro, Dongdaemun-gu, Seoul 02447, Korea; conan2010@naver.com

**Keywords:** organizing pneumonia, pneumonitis, vancomycin

## Abstract

The long-term administration of vancomycin has increased; however, the pulmonary adverse reactions of long-term vancomycin treatment remain under-studied. A 75-year-old male patient with vertebral osteomyelitis receiving long-term vancomycin therapy developed a fever. High resolution computed tomography showed irregular ground glass opacity and consolidation in the right upper lung. The patient developed organizing pneumonia. This occurred without peripheral eosinophilia or adverse reactions in the skin and liver. The administration of vancomycin was discontinued. He recovered from organizing pneumonia after four weeks of steroid therapy. Solitary organizing pneumonia can develop during treatment with vancomycin. When pulmonary inflammation occurs and other causes of pneumonia are excluded, vancomycin therapy should be discontinued.

## 1. Introduction

Vancomycin is a tricyclic glycopeptide antibiotic that is used as a first line agent for many infections, including bacteraemia, endocarditis, pneumonia, cellulitis, osteomyelitis, caused by methicillin-resistant *Staphylococcus aureus,* and coagulase-negative Staphylococcus infections [1]. This drug is associated with several adverse drug reactions such as nephrotoxicity, infusion-related events, ototoxicity, and hypersensitivity reactions [1]. As the incidence of vertebral osteomyelitis, primarily caused by *S. aureus* increases, long-term administration of vancomycin also increases [2]. However, the adverse effects of long-term vancomycin treatment remain under-studied.

Previous case reports have shown that pulmonary adverse effects may occur with vancomycin-induced drug reaction with eosinophilia and systemic symptoms (DRESS) syndrome [3]. Herein, we describe a case of organizing pneumonia caused by long-term treatment with vancomycin. This study was approved by the Institutional Review Board (KHUH-2020-07-039; 16 July 2020) of Kyung Hee University Hospital, Seoul, Korea, which waived the need for written informed consent from the patient.

## 2. Case Report

A 75-year-old male patient came to the emergency room of Kyung Hee Medical Centre due to back pain and fever. The patient takes losartan (50 mg/day) for hypertension, metformin (500 mg/day) for diabetes, and tamsulosin (0.2 mg/day) and alfuzosin (10 mg/day) for benign prostatic hyperplasia. The patient did not take any immunosuppressive agents before admission. He was a non-smoker and did not consume alcohol regularly. The patient underwent surgery for rectal cancer 10 years ago. He had received surgery for spinal stenosis 6 years ago. The patient had a fever of up to 38.1 °C. In laboratory examinations, white blood cell count was 14.71 × 10^9^/L (98% neutrophil) and C-reactive protein was 318.8 nmol/L. After magnetic resonance imaging of the lumbar spine, he was diagnosed with vertebral osteomyelitis at the L3-L4 lumbar spine level. He underwent anterior decompression surgery on the third hospital day and posterior debridement and posterior instrumentation surgery on the tenth hospital day. The pus and infected tissues were obtained for ordinary culture during surgery, but the pathogen was not proven by the culture. The patient was prescribed the intravenous vancomycin (1.6 g per day) for 41 days and meropenem (3 g per day) on first day of hospitalization. The treatment with meropenem was stopped on the 28th hospital day. On the 42nd hospital day, the patient developed a fever of 39.6 °C. In laboratory examinations, the white blood cell count was 19.81 × 10^9^/L (98% neutrophil) and the C-reactive protein was 245.7 nmol/L. Serum AST and ALT were near normal at 57 and 21 IU/L, and serum total bilirubin and direct bilirubin levels were 2.73 and 1.72 mg/dL. Serum BUN and creatinine concentrations were 27 mg/dL and 0.91 mg/dL. Serum activated partial thromboplastin time (aPTT) value was 45.7 s (normal range, 29–43 s). His serum sodium and potassium levels were 133 and 3.7 mmol/L. A chest X-ray showed increased opacity in the right upper lobe (Figure 1A). High resolution computed tomography (HRCT) showed irregular ground glass opacity and consolidation in the right upper lung (Figure 1B). There was no visible endobronchial lesion, and the purulent discharge was not observed in the area of the tracheal carina and main bronchi on bronchoscopy. There was no evidence of alveolar haemorrhage through bronchoscopy. Considering hospital-acquired pneumonia or vancomycin-induced pneumonitis as the probable cause of lung lesion, he received meropenem again, and the administration of vancomycin was discontinued 8 days after the onset of fever. The intermittent fever was sustained for 9 days, and the patient became apyretic on day 14 after fever onset, with a normal white blood cell count and C-reactive protein. Despite clinical improvement, the right upper lung lesion following a chest X-ray was not improved. Gram stain and ordinary culture from a sputum and polymerase chain reaction for *Mycoplasma pneumoniae*, *Legionella pneumoniae*, and *Chlamydia pneumoniae* in respiratory specimens revealed no pathologic organisms. Percutaneous needle biopsy or biopsy by video-assisted thoracic surgery were considered, but the invasive examination could not be performed due to the consideration of the patient’s age and general condition. A comprehensive diagnosis of organizing pneumonia was made based on the clinical course and the results of the imaging tests. On the 60th day of hospitalization, he was prescribed a daily 30 mg dose of prednisolone for 4 weeks. The medications for underlying diseases, including hypertension, diabetes, and benign prostatic hyperplasia, were administered without cessation during the entire hospitalization. The right upper lung lesion disappeared on the chest X-ray (Figure 1C). He had no symptoms or signs relating to malignancy and autoimmune diseases until 2 months after the onset of organizing pneumonia.

## 3. Discussion

Our patient developed alveolar opacity in the right upper lobe after 6 weeks of administration of vancomycin for the treatment of vertebral osteomyelitis. Because the radiologic findings and the clinical course improved after the withdrawal of the vancomycin and the administration of corticosteroid, it was assumed that organizing pneumonia occurred as a result of vancomycin administration. This is the first report regarding the development of solitary organizing pneumonia due to vancomycin administration.

Organizing pneumonia belongs to a diffuse interstitial lung disease in which damage to the alveolar wall occurs [4]. There are several associated factors of organizing pneumonia, such as infection, connective tissue disease, drugs, immunodeficiency, and malignancy [4]. Drugs are a major cause of organizing pneumonia [4]. More than 20 drugs are associated with organizing pneumonia. Among them, antibiotics, nitrofurantoin, minocycline, amphotericin B, and cephalosporin are known to be related to organizing pneumonia. In addition, a case of organizing pneumonia and pulmonary eosinophilic infiltration after daptomycin administration for 4 weeks was reported [5]. However, the development of organizing pneumonia caused by vancomycin administration was not known until recently. The exact pathogenesis of organizing pneumonia is uncertain. It is presumed that alveolar epithelial injury due to certain drugs, and sequential inflammation in the alveolar lumen may be the main pathway of organizing pneumonia [6]. It is difficult to determine whether the drug is responsible, since organizing pneumonia may be associated with other underlying conditions. A diagnosis of organizing pneumonia can be made by the typical histopathologic findings of biopsied lung tissue in patients with compatible clinical and radiological features [4]. This method is also required to exclude other aetiologies of organizing pneumonia. In this case study, the medications for underlying diseases could be ruled out as the causes of organizing pneumonia, as medications for the treatment of underlying diseases were administered without cessation during the entire hospitalization period. The patient had no symptoms or signs relating to malignancy and autoimmune diseases until 2 months after the onset of organizing pneumonia. The advanced age and systemic conditions of the case patient hindered the confirmative diagnosis of organizing pneumonia by lung tissue biopsy. However, the lung lesion of our patient was presumed to be organizing pneumonia based on comprehensive clinical and radiological findings, excluding other settings. An evaluation of the causative agents is possible by identifying the improvement of the lung lesions after discontinuation of the drug.

Generally, pulmonary side effect due to administration of vancomycin is caused by the hypersensitivity reactions [7]. Skin lesions are usually presented as the initial symptom, and are often accompanied by liver and kidney injuries [3]. Pulmonary involvement is estimated as 5% in DRESS syndrome [8,9]. Pulmonary side effects due to vancomycin administration according to drug-induced hypersensitivity and eosinophilic pneumonia are shown in Table 1. Wilcox et al. summarized 23 cases of vancomycin-induced DRESS syndrome from 1997 to 2016, of which 4 cases showed lung involvement [8]. They also reported a case of a 39-year-old man with osteomyelitis of the foot, who was diagnosed with DRESS syndrome with pulmonary manifestation as an initial symptom, after the administration of vancomycin for 3 weeks. Kwon et al. reported hypersensitivity due to vancomycin therapy in a patient with vertebral osteomyelitis [7]. The patient suffered from a rash, fever, peripheral eosinophilia, interstitial pneumonitis, and interstitial nephritis, following the administration of vancomycin and teicoplanin. The cases of eosinophilic pneumonia due to vancomycin administration were reported. It is estimated that eosinophilic pneumonia is caused by a hypersensitivity reaction due to inhaled antigens, but the exact mechanism is still unclear. Isono et al. reported a case of acute eosinophilic pneumonia, confirmed by bronchoalveolar lavage after administration of intravenous vancomycin for 34 days in a 65-year-old man who was diagnosed with MRSA empyema [10]. Peripheral eosinophilia was also present. Compared with the above-mentioned cases, liver and skin involvement did not occur in our patient, and peripheral eosinophilia was also absent. Solitary pulmonary adverse reactions manifested as organizing pneumonia developed during the long-term administration of vancomycin.

## 4. Conclusions

We reported the development of solitary organizing pneumonia in a patient with vertebral osteomyelitis who was treated with vancomycin for about 6 weeks. Pulmonary adverse reactions due to vancomycin therapy is uncommon. However, when pulmonary inflammation occurs during vancomycin administration, vancomycin should be considered as the offending agent of the lungs, but only after other cases of pneumonia have been excluded.

## Figures and Tables

**Figure 1 medicina-57-00610-f001:**
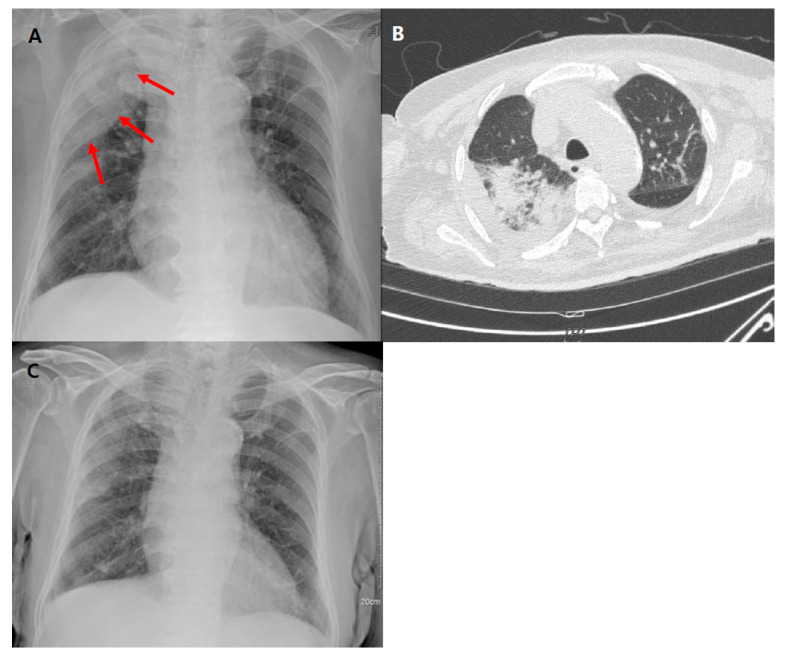
Chest X-ray and high-resolution computed tomography (HRCT) of a case patient. (**A**) Chest X-ray showed increased opacity in the right upper lobe. (**B**) HRCT showed irregular ground glass opacity and consolidation in the right upper lung. (**C**) Chest X-ray showed that the right upper lung lesion had disappeared.

**Table 1 medicina-57-00610-t001:** Drug-induced hypersensitivity and eosinophilic pneumonia cases associated with adverse pulmonary reactions and vancomycin administration.

	Drug-Induced Hypersensitivity					Eosinophilic Pneumonia
Patient (Age/Sex)	79/Male	50/Male	66/Male	57/Male	38/Female	65/Male
Relevant medical history	None	Alcoholic liver cirrhosis	Heterozygous hemochromatosis	Congenital heart disease (Aortic, pulmonary valve replacement)	None	IgA nephropathy
Infectious disease	Wound infection after femur fracture	Vertebral osteomyelitis with epidural abscess	Implant infection at pelvis	Infective endocarditis	Infective endocarditis	Empyema with pneumothorax
Microorganism	Methicillin resistant *Staphylcoccus aureus*	-	Methicillin resistant *Staphylcoccus aureus*	Penicillin-resistant *Staphylococcus epidermidis*	*Streptococcus oralis*	Methicillin-resistant *Staphylcoccus aureus*
History of antibiotics	Vancomycin for 29 days then, Teicoplanin was initiated	Vancomycin for 18 days, then switched to Ceftriaxone, then switched to Teicoplanin	Vancomycin for 4 weeks	Vancomycin for 4 weeks	Amoxicillin, Gentamicin, and Vancomycin for 3 weeks Rifampicin, Teicoplanin	Vancomycin for 2 days Piperacillin/tazobactam Meropenem
Involved organs	Skin, Lung	Skin, Lung	Skin, Lung, Liver	Skin, Lung, Gastrointestinal tract	Skin, Lung, Kidney	Lung
Clinical manifestations	Fever Eosinophilia Skin rash Diffuse pneumonic infiltrates	Skin rash High fever up to 39 °C Eosinophilia Hypersensitivity pneumonitis	Skin rash High fever up to 40 °C Cervical lymphadenopathy Eosinophilia Abnormal liver enzymes Eosinophilic pneumonitis	Fever Epigastric pain Diarrhoea Maculopapular pruritic rash Severe hypoxia Peripheral eosinophilia Acalculous cholecystitis	Fever Upper body erythema Renal failure with dialysis Mechanical ventilation	Peripheral eosinophilia Eosinophil dominant in bronchoalveolar lavage fluid
Treatment	Prednisolone 50 mg for 6 days Switched to linezolid	Methylprednisolone 30 mg q 6 h Stop antibiotics Switched to linezolid	Stop antibiotics Topical steroid with antihistamine Prednisolone 60 mg	Prednisolone 40 mg	Methylprednisolone 1 mg/kg/day	Switched to linezolid Prednisolone 30 mg
Outcome	Resolved	Resolved	Resolved	Resolved	Resolved	Resolved
Reference	DRESS syndrome caused by cross reactivity between vancomycin and subsequent teicoplanin administration [9]	A Case of Hypersensitivity syndrome to both vancomycin and teicoplanin [7]	DRESS with delayed onset acute interstitial nephritis and profound refractory eosinophilia secondary to vancomycin [11]	Vancomycin induced DRESS syndrome in a patient with tricuspid endocarditis [12]	Severe vancomycin-induced drug rash with eosinophilia and systemic symptoms syndrome imitating septic shock [13]	Eosinophilic pneumonia putatively induced by vancomycin: a case report [10]

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
