# Peer review of "Vancomycin-Induced Organizing Pneumonia: A Case Report and Literature Review"

_medicina, 2021, doi:10.3390/medicina57060610_

Round 1

Reviewer 1 Report

The manuscript has been much improved. I have no further comments. 

Reviewer 2 Report

The authors have appended their manuscript and included details on patient medication and laboratory tests undertaken.

While rare, this pathology may be of interest as a reference to other clinicians and scientists treating/working on OP.

This manuscript is a resubmission of an earlier submission. The following is a list of the peer review reports and author responses from that submission.

Round 1

Reviewer 1 Report

It is an interesting case report and it sounds plausible that vancomycin may be the cause of organizing pneumonia. However, there are a number of reasons for this diagnosis, and I can doubt whether the authors have ruled out all causes: i.a. malignancy, other medications, immunodeficiency.

In addition, some clinical information about this patient is missing, i.e. tobacco consumption and alcohol consumption which may be contributing factors to poor immune response 
Last,  it is also important with other biochemical information, i.e., liver bloodtests, kidney counts, electrolytes and coagulation 

Author Response

  1. It is an interesting case report and it sounds plausible that vancomycin may be the cause of organizing pneumonia. However, there are a number of reasons for this diagnosis, and I can doubt whether the authors have ruled out all causes: i.a. malignancy, other medications, immunodeficiency. --> I would like to thank you for the valuable and helpful comments on our submitted manuscript.  We have ruled out the above-mentioned possible causes of organizing pneumonia. 1) Medication The patient has taken losartan (50 mg/day) for hypertension, metformin (500 mg/day) for diabetes, and tamsulosin (0.2 mg/day) and alfuzosin (10 mg/day) for benign prostatic hyperplasia before admission. These medications administered without cessation during the entire hospitalization. Therefore, these medications may not have an impact on the organizing pneumonia. 2) MalignancyThe patient underwent surgery for rectal cancer 10 years ago, and there was no evidence of recurrence of rectal cancer. In addition, the patient had no symptom or sign related to malignancy in other organs until 2 months after the onset of organizing pneumonia. 3) Autoimmune disease and immunodeficiency There was no evidence of autoimmune disease. Serologic markers for autoimmune diseases, including anti-neutrophil cytoplasmic antibody (ANCA), rheumatoid factor, and anti-CCP antibody, were all negative. The ANA was positive with a titer of 1:40. However, this result considered as the false-positive due to the advanced age, because no symptom or sign associated with autoimmune diseases was not noted before and after developing organizing pneumonia. A human immunodeficiency virus antibody result was negative. The patients did not take any immunosuppressive agents before this event. As the reviewer’s comment, we added these points in the manuscript as follows.  “The patient has taken losartan (50 mg/day) for hypertension, metformin (500 mg/day) for diabetes, and tamsulosin (0.2 mg/day) and alfuzosin (10 mg/day) for benign prostatic hyperplasia. The patients did not take any immunosuppressive agents before admission.” “The patient underwent surgery for rectal cancer 10 years ago.” “The medications for underlying diseases, including hypertension, diabetes, and benign prostatic hyperplasia, administered without cessation during the entire hospitalization.” “He had no symptom or sign related to malignancy and autoimmune diseases until 2 months after the onset of organizing pneumonia.” Discussion section:“It is also needed to exclude other etiology of organizing pneumonia. In this case, the medications for underlying diseases could be ruled out as the causes of organizing pneumonia, because the medications for underlying diseases administered without cessation during the entire hospitalization. He had no symptom or sign related to malignancy and auto-immune diseases until 2 months after the onset of organizing pneumonia.” 
  2. In addition, some clinical information about this patient is missing, i.e. tobacco consumption and alcohol consumption which may be contributing factors to poor immune response. --> The patient was non-smoker, and did not consume alcohol regularly. As review’s comment, we added these information in the manuscript as follows.  “He was non-smoker, and did not consume alcohol regularly.”  
  3. Last,  it is also important with other biochemical information, i.e., liver bloodtests, kidney counts, electrolytes and coagulation. --> As review’s comment, we added these information in the manuscript as follows. “Serum AST and ALT were near normal at 57 and 21 IU/L, and serum total bilirubin and direct bilirubin levels were 2.73 and 1.72 mg/dL. Serum BUN and creatinine concentrations were 27 mg/dL and 0.91 mg/dL. Serum activated partial thromboplastin time (aPTT) value was 45.7 seconds (normal range, 29-43 seconds). His serum sodium and potassium levels were 133 and 3.7 mmol/L.”

Reviewer 2 Report

The presentation is of potential interest to pulmonologists, internal physicians, infectious disease' physicians and other medical specialities treating with vancomycin.

The manuscript would be improved by addition of bronchoscopy (cytology) data, if available.

Review of English needs to be done for it to adhere to norms.

Author Response

The presentation is of potential interest to pulmonologists, internal physicians, infectious disease' physicians and other medical specialities treating with vancomycin.The manuscript would be improved by addition of bronchoscopy (cytology) data, if available. Review of English needs to be done for it to adhere to norms.

-->I would like to thank you for the valuable and helpful comments on our submitted manuscript. Bronchoscopy was performed for evaluation of the cause of pulmonary lesion. There was no visible endobronchial lesion and the purulent discharge was not observed in the area of the tracheal carina and main bronchi on bronchoscopy. We could rule out the possibility of alveolar hemorrhage through bronchoscopy. Unfortunately, cytology test with bronchial washing specimens was not performed. Gram stain and ordinary culture from sputum and polymerase chain reaction for Mycoplasma pneumoniae, Legionella pneumoniae, and Chlamydia pneumoniae in respiratory specimens revealed no pathologic organisms. As the reviewer’s comment, we added this result in the manuscript as follows.  

“There was no visible endobronchial lesion and the purulent discharge was not observed in the area of the tracheal carina and main bronchi on bronchoscopy. There was no evidence of alveolar hemorrhage through bronchoscopy.” 

“Gram stain and ordinary culture from sputum and polymerase chain reaction for Mycoplasma pneumoniae, Legionella pneumoniae, and Chlamydia pneumoniae in respiratory specimens revealed no pathologic organisms.”  

We believe we have addressed all questions and comments in a suitable fashion, but would be happy to provide further information or revision if necessary.

Sincerely yours,

Yu-Mi Lee, M.D.

Division of Infectious Diseases,

Department of Internal Medicine

Kyung Hee University School of Medicine

23, Kyungheedae-ro, Dongdaemun-gu,

Seoul, 02447, Republic of Korea

Tel: 82-2-958-8209, Fax: 82-2-968-1848

E-mail: cristal156@hanmail.net
